# The Short-Term Effects of Ambient Air Pollutants on Childhood Asthma Hospitalization in Taiwan: A National Study

**DOI:** 10.3390/ijerph16020203

**Published:** 2019-01-12

**Authors:** Ching-Yen Kuo, Chin-Kan Chan, Chiung-Yi Wu, Dinh-Van Phan, Chien-Lung Chan

**Affiliations:** 1Department of Information Management, Yuan Ze University, 135 Yuan-Tung Road, Jung-Li, Taoyuan 320, Taiwan; jasmine@mail.tygh.gov.tw (C.-Y.K.); johnny82128@gmail.com (C.-Y.W.); dvan2707@due.edu.VN (D.-V.P.); 2Department of Medical Administration, Taoyuan General Hospital, Ministry of Health and Welfare, 1492 Zhongshan Road, Taoyuan Dist., Taoyuan 330, Taiwan; 3Department of Pediatrics, Taoyuan General Hospital, Ministry of Health and Welfare, 1492 Zhongshan Road, Taoyuan Dist., Taoyuan 330, Taiwan; jean620104@yahoo.com.tw; 4Department of Biotechnology, Ming Chuan University, 5 De Ming Road, Gui Shan Dist., Taoyuan 333, Taiwan; 5Innovation Center for Big Data and Digital Convergence, Yuan Ze University, 135 Yuan-Tung Road, Jung-Li, Taoyuan 320, Taiwan; 6University of Economics, The University of Danang, 71 Ngu Hanh Son Street, Danang 550000, Vietnam

**Keywords:** children, air pollution exposure, asthma, hospitalization

## Abstract

This investigation determined the effects of air pollution on childhood asthma hospitalization in regions with differing air pollution levels in Taiwan over a long time period. Data of childhood hospital admissions for asthma in patients aged 0–18 years and air quality in eight regions for the period 2001–2012 in Taiwan were collected. Poisson generalized linear regression analysis was employed to identify the relative risks of hospitalization due to asthma in children associated with exposure to varying levels of air pollutants with a change in the interquartile range after adjusting for temperature and relative humidity. Particulate matter ≤2.5 μm (PM_2.5_), particulate matter ≤10 μm (PM_10_), ozone (O_3_), sulfur dioxide (SO_2_), and nitrogen dioxide (NO_2_), were positively associated with childhood asthma hospitalization, while O_3_ was negatively associated with childhood asthma hospitalization. SO_2_ was identified as the most significant risk factor. The relative risks for asthma hospitalization associated with air pollutants were higher among children aged 0–5 years than aged 6–18 years and were higher among males than females. The effects of air pollution on childhood asthma were greater in the higher-level air pollution regions, while no association was observed in the lower-level air pollution regions. These findings may prove important for policymakers involved in implementing policies to reduce air pollution.

## 1. Introduction

Asthma is a chronic inflammatory airway disease, characterized by coughing, wheezing, dyspnea and chest tightness, that may originate in early life [1]; it is a global problem affecting approximately 10% of children worldwide, with huge individual and societal burden. Asthma is one of the major reasons for school absence, emergency medical treatment, and hospitalization during childhood [2]. Asthma hospitalization represents a serious adverse outcome that is theoretically preventable with high-quality healthcare, patient education, and optimal treatment [3]. Many environmental factors have been linked to asthma causation; obesity, urban living, dietary pattern, cigarette smoking, air pollution, and viral infections are all associated with asthma exacerbation in children. Air pollution and its related health impact have become a major concern over the past few years. Ambient air pollution accounts for an estimated 4.2 million deaths per year due to stroke, heart disease, lung cancer, and chronic respiratory diseases. It has been estimated that 91% of the world’s population lives in places where air quality exceeds WHO guideline limits. The pollutants with the strongest evidence of health effects are particulate matter ≤2.5 μm (PM_2.5_), particulate matter ≤10 μm (PM_10_), ozone (O_3_), sulfur dioxide (SO_2_), and nitrogen dioxide (NO_2_) [4]. Ambient air pollution levels have been found to be associated with hospitalization due to asthma [5]. A systematic review and meta-analysis of the association between air pollutants and asthma hospital admissions showed that air pollutants were associated with a significant increased risk of asthma hospitalizations [6].

The major mechanisms of individual air pollutants responsible for triggering asthma exacerbation are thought to be associated with oxidative injury to the airways, leading to inflammation, remodeling, and an increased risk of sensitization. However, various environmental and weather conditions could have different effects in different regions. To the best of our knowledge, no previous study has examined the impact of air pollution on childhood asthma hospitalization in different regions of Taiwan at the national level over a long duration with large sample sizes. Therefore, we integrated data from the National Health Insurance Research Database (NHIRD) from 2001 to 2012 with air pollution and weather data from Taiwan governmental open data sets in order to assess the effects of air pollution in regions of differing levels of air pollution on childhood asthma hospitalization.

## 2. Materials and Methods

### 2.1. Asthma Hospitalization Data

Childhood asthma hospitalization data were obtained from the National Health Insurance Research Database (NHIRD) established by the National Health Insurance Administration, Ministry of Health and Welfare, Taiwan. Taiwan launched a single-payer National Health Insurance program on March 1, 1995. As of 2014, 99.9% of Taiwan’s population was enrolled. The database of this program contains registration files and original claim data for reimbursement, and is maintained by the National Health Research Institutes (NHRI), Taiwan [7]. The NHIRD includes various data subsets, such as inpatient expenditure by admission (DD), details of inpatient orders (DO), ambulatory care expenditure by visit (CD), and details of ambulatory care orders (OO). In this study, we used the inpatient expenditure by admission DD data subset from 2001 to 2012, cases being identified when the International Classification of Diseases, Ninth Revision, Clinical Modification (ICD-9-CM) code for asthma (493.XX) was listed as the major diagnosis in children under the age of 18. Data were managed using the Impala Hadoop big data management system and retrieved via the RDBMS (Relational Database Management System).

### 2.2. Air Pollution and Meteorological Data

Data on levels of air pollutants were obtained from Taiwanese Environmental Protection Administration (EPA) air quality monitoring stations. The stations are reasonably distributed in counties and cities. The data are also believed to be representative of the region. Taiwan is divided into seven air quality regions and an outlying islands region according to the pollution characteristics, topography and weather conditions by the EPA. The range of each region and the number of stations are as follows.

In Northern Taiwan, there are 26 stations; in the Chu-Miao region, 6 stations; in Central Taiwan, 11 stations; in the Yun-Chia-Nan region, 11 stations; in the Kao-Ping region, 16 stations; in the Yilan region, 2 stations; and in the Hua-Tung region, 3 stations. In the outlying islands region (Matsu, Kinmen, Magong), there are three stations.

Asthma patients’ addresses were not available from the database, and we therefore assumed that a patient’s area of residence was close to the location of the hospital to which they were admitted. Based on their locations, hospitals were divided into the seven air quality regions and the outlying islands region according to the EPA regions. We selected air pollutant monitoring stations located in the same administrative division as the hospital to which patients were admitted. Each station takes hourly measurements of air pollutants, giving 24-h average daily concentrations for the following pollutants: PM_2.5_, PM_10_, O_3_, SO_2_, and NO_2_. Daily air pollution concentrations were averaged from the available monitoring data of stations located in the 8 regions. When data were missing for a particular monitoring station on a given day, the measurements recorded at other nearest monitoring stations were used to calculate the daily average values. It is well documented that temperature and humidity can be potential confounders by other studies, so the ambient daily temperature and relative humidity were used to control for meteorological conditions. Meteorological data including the ambient daily average temperature and relative humidity were obtained from the Central Weather Bureau. The air quality regions of Taiwan are shown in Figure 1.

### 2.3. Statistical Analysis

As the number of daily hospital admissions due to asthma is a type of small probability event, and typically follows a Poisson distribution [8]. Single-pollutant model aimed at estimating the increased risk of adverse health outcomes, associated with the exposure to a single air pollutant. However, air pollution is a mixture of many different gases, vapors and particles, with varying concentration and composition depending on the geographic regions and meteorological conditions. Because humans are simultaneously exposed to a complex mixture of air pollutants, many organizations have encouraged moving towards a multi-pollutant model to air quality [9,10]. In this study, we applied univariate and multivariate Poisson generalized linear regression models were used to investigate the short-term associations between daily childhood asthma hospitalizations and daily air pollution. The dependent variables were the daily numbers of childhood asthma hospitalizations, while the independent variables were the 24-h average daily concentrations of PM_2.5_, PM_10_, O_3_, SO_2_, and NO_2_.

Results are presented as relative risks (RRs) and 95% confidence intervals (CIs) associated with an interquartile range (IQR) increase in the levels of PM_2.5_, PM_10_, O_3_, SO_2_, and NO_2_ after adjusting for the ambient daily temperature and relative humidity. All tests were conducted at a significance alfa level of 0.05. We also explored the effects of air pollution on childhood asthma hospitalization with regards to patient age and gender. All models were performed using R software version 3.3.2 (The R Foundation for Statistical Computing, Vienna, Austria).

## 3. Results

Table 1 presents the demographic characteristics of the patients, seasonal case distribution and hospital admissions per region due to asthma from 2001 to 2012. During the study period, there were 59,204 hospitalizations due to asthma in children aged 0–18 years in Taiwan. The number of cases was highest among those aged 3–5 years, accounting for 25,043 patients (42.30%), followed by 6–18 years, accounting for 22,779 patients (38.48%). There were more hospitalizations due to asthma of male patients (37,825 patients; 63.89%) than female patients (21,379 patients; 36.11%). In terms of seasonal distribution, asthma hospitalization occurred most often in winter (32.49%) and spring (26.89%). The average length of hospital stay during the study was 3.93 bed-days in patients aged 0–18 years. The lowest average length of stay was observed in the Northern region, which was 3.54 bed-days, while the highest was in the Hua-Tung region, which was 4.51 bed-days. The region with the most hospitalizations due to asthma in children was the Northern region (37.97%), followed by Central Taiwan (18.95%), with the lowest number of cases being observed in the outlying islands region (0.56%).

Table 2 shows the daily mean concentrations of ambient air pollutants. The air pollutant levels differed significantly between regions. The concentrations of PM_2.5_, PM_10_, and SO_2_ during 2001–2012 in the Kao-Ping region were higher than those in the other regions; this was not the case for O_3_ and NO_2_. The level of air pollution was lower in the Hua-Tung region. The daily mean concentration of O_3_ was highest in the outlying islands region, while that of NO_2_ was highest in the Northern region.

Table 3 summarizes the relative risk of hospitalization (RR) due to asthma and air pollution with a change in the interquartile range (IQR) of the level of the pollutant exposed to after adjusting for the meteorological factors of temperature and humidity in single pollutant and multiple pollutants model. Among children under 18 years of age, we observed significant increases in asthma hospitalization with interquartile range increases in PM_2.5_ (RR = 1.156; CI = 1.142–1.170; *p* < 0.001), PM_10_ (RR = 1.120; CI = 1.107–1.134; *p* < 0.001), SO_2_ (RR = 1.367; CI = 1.349–1.385; *p* < 0.001) and NO_2_ (RR = 1.065; CI = 1.053–1.076; *p* < 0.001), with the strongest effect estimate being observed for SO_2_, while O_3_ (RR = 0.969; CI = 0.957–0.981; *p* < 0.001) was negatively associated with asthma hospitalization in the univariate Poisson regression. We also found that all pollutants were associated with asthma hospitalization in the multivariate Poisson regression after adjusting for all meteorological factors and pollutants. The association with hospitalization remained strongest for SO_2_ among all children (RR = 1.537; CI = 1.507–1.566; *p* < 0.001). This association persisted when patients were divided into groups by age and gender.

According to age and gender stratified analysis (Table 4), the RRs for asthma hospitalization were higher among children aged 0–5 than in children aged 6–18 in the single pollutant model. The RR for an IQR increase in SO_2_ was highest among children aged 0–5 years (RR = 1.428; CI = 1.405–1.452; *p* < 0.001) and clearly lower in those aged 6–18 years (RR = 1.245; CI = 1.219–1.271; *p* < 0.001). Similar patterns were observed for PM_2.5_ and PM_10_. In the 0–5 years group, PM_2.5_ (RR = 1.190; CI = 1.172–1.208; *p* < 0.001) and PM_10_ (RR = 1.123; CI = 1.106–1.140; *p* < 0.001) were associated with childhood asthma hospitalization, while in the 6–18 years group, PM_2.5_ (RR = 1.088; CI = 1.067–1.109; *p* < 0.001) and PM_10_ (RR = 1.100; CI = 1.078–1.121; *p* < 0.001) were less strongly associated with childhood asthma hospitalization than in children aged 0–5 years. In the multivariate Poisson regression, we observed significant relationships between O_3_, SO_2_ and NO_2_ and asthma hospitalization in the 0–5 years age group, and all pollutants were associated with asthma hospitalization in the 6–18 years group. Of all pollutants, SO_2_ was most strongly associated with daily asthma hospitalization among children aged 0–5 years (RR = 1.647; CI = 1.607–1.689; *p* < 0.001) and those aged 6–18 years (RR = 1.346; CI = 1.306–1.387; *p* < 0.001). In terms of gender, we found that all pollutants were associated with asthma hospitalization in both males and females in the univariate Poisson regression. The RRs for asthma hospitalization were slightly higher among males than females. In the multivariate Poisson regression, we identified significant relationships between air pollutants and asthma hospitalization, the associations being strongest with SO_2_ among both the male (RR = 1.525; CI = 1.489–1.562; *p* < 0.001) and female patients (RR = 1.495; CI = 1.448–1.543; *p* < 0.001).

Table 5 shows the effect estimates of air pollutants on childhood asthma hospitalization in each region in a single pollutant model. We observed some associations between air pollutants and childhood asthma hospitalization in only some regions. In the univariate Poisson regression, PM_2.5_ and PM_10_ were only associated with childhood asthma hospitalization in the Northern, Central Taiwan, Yun-Chia-Nan, and Kao-Ping regions. O_3_ was only negatively associated with childhood asthma hospitalization in Central Taiwan. SO_2_ was associated with childhood asthma hospitalization in the Northern region, Central Taiwan, the Yun-Chia-Nan region, and the Kao-Ping region. NO_2_ was associated with childhood asthma hospitalization in the Yun-Chia-Nan and the Kao-Ping region.

Table 6 shows the effect estimates of air pollutants on childhood asthma hospitalization in each region in multiple pollutants model. PM_2.5_ and SO_2_ were positively associated with childhood asthma hospitalization but O_3_ and NO_2_ was negatively associated with childhood asthma hospitalization in the Northern region. PM_10_ and SO_2_ were positively associated with childhood asthma hospitalization but PM_2.5_ and NO_2_ was negatively associated with childhood asthma hospitalization in the Central Taiwan. In the Yun-Chia-Nan and the Kao-Ping region, only SO_2_ was positively associated with childhood asthma hospitalization .There was a clear indication that higher SO_2_ levels were associated with increased numbers of hospitalizations for childhood asthma in the Northern region (RR = 1.193; CI = 1.167–1.219), Central Taiwan (RR = 1.347; CI = 1.292–1.404) , Yun-Chia-Nan (RR = 1.178; CI = 1.092–1.271), and the Kao-Ping region (RR = 1.172; CI = 1.086–1.265).

## 4. Discussion

This study compared the effect of exposure to air pollution on hospitalization due to childhood asthma in different regions with different air pollution patterns in Taiwan, and the results showed consistent and statistically significant increases in the RRs for asthma hospitalization under increased levels of air pollution in different age and gender groups. We also identified differing associations between asthma hospitalization in children and air pollution levels in different regions of Taiwan. It is important to note that of all the pollutants examined, SO_2_ was most strongly associated with daily asthma hospitalizations in children. The results were generally consistent with other studies showing that hospital admissions for childhood asthma are associated with air pollution.

In a study based on Poisson regression, the relevance of exposure to PM_2.5_ to hospitalizations due to pneumonia, acute bronchitis, bronchiolitis, and asthma among people living in Volta Redonda was shown. An increase in the PM_2.5_ concentration being found to result in significant increase of up to 9 percentage points in the risk of hospitalization due to pneumonia, acute bronchitis, bronchiolitis and asthma [11]. In another study based on Poisson regression, it was found that a 10 μg/m^3^ increase in PM_10_ was associated with a 2.54% increase in the number of pediatric asthma hospital admissions [12]. In Nhung et al.’s study, the PM_10_ concentration had effects on hospital admissions with a two-day lag for respiratory diseases in children under 15 years of age [13]. Another study by Amâncio and Nascimento demonstrated that an increase in PM_10_ of 17 μg/m^3^ resulted in an increase in the RR of 16% for hospitalization due to asthma [14].

PM originates from the combustion process of diesel and gasoline-powered vehicles, burning of biomass and burning of coal to generate power. PM is a complex mixture of solid and liquid particles suspended in air. The size, chemical composition, and other physical and biological properties of particles vary with location and time. This heterogeneity in PM components may cause different health effects through various pathways [15,16], and it has been suggested that there is a degree of heterogeneity in the effect of particulate matter on mortality within the same country [17].

O_3_ was negatively associated with the daily numbers of childhood asthma hospitalizations after adjusting for temperature and humidity in our study. In contrast to other studies, no consistent association between childhood asthma hospitalization and O_3_ was observed in this study. Samoli et al. [12] showed that O_3_ exposure was associated with a statistically significant increase in asthma admissions among older children in the summer. According to a Poisson regression analysis, Nhung et al. [13] did not find a statistically significant association between O_3_ exposure and the daily number of hospitalizations for asthma. According to a review of 87 studies [6], O_3_ was found to be significantly associated with an increased risk of asthma-related hospitalization in 71 studies. In a nationwide cross sectional study about the effect of air pollutants on the risk of asthma among school children in 2001 in Taiwan [18], Hwang et al. found the risk of childhood asthma was positively associated with O_3_ (adjusted OR 1.138, 95% confidence interval 1.001 to 1.293), The level of O_3_ is affected by sunlight, temperature and other air pollutants; increased sunlight and temperatures increase the production of tropospheric ozone due to the photochemical nature of the secondary pollutant. The relationship between the O_3_ level and childhood asthma hospitalization requires further research.

In our study, the level of SO_2_ was most strongly associated with daily asthma hospitalizations among children of all the pollutants studied. SO_2_ was found to be associated with hospitalization due to asthma in the Northern region, Central Taiwan, the Yun-Chia-Nan region, and the Kao-Ping region, and was the major pollutant affecting asthma hospitalization in these regions. A study that employed Poisson regression analysis was conducted in Brazil, and showed that an increase in the concentration of SO_2_ of 3 μg/m^3^ led to an increase in the RR of hospitalization due to asthma of 8% [14], while the same increase in SO_2_ was associated with a 5.98% increase in the RR [12]. Zheng et al. [6] reviewed 87 studies focused on air pollutants and asthma-related emergency room visits and hospitalizations, and 65 studies demonstrated a statistically significant correlation between asthma exacerbation and the level of SO_2_. Another systematic review showed that SO_2_ was significantly associated with asthma exacerbation in children aged 0–18 [19]. However, the risk of childhood asthma was not related to SO_2_ (adjusted OR 0.874, 95% CI 0.729 to 1.054) in 2001 in Taiwan by Hwang et al. [18]. Because urban air pollution constitutes a complex mixture of several compounds, SO_2_ and PM_10_ concentrations were also correlated (Hwang et al.), assessment of the independent effects of different pollutants is difficult [18].

The main sources of SO_2_ in the developed world are primary emissions during energy production or industrial processes [20]. SO_2_ is a recognized environmental toxicant that can act to promote airway responses in a concentration-dependent manner, possibly through its ability to induce local oxidative stress [21]. A high probability of SO_2_ exposure may be confined to the factory area itself and within the vicinity of several square miles or the original site of its generation [22]. A study conducted in Russia by Nieminen et al. (2013) sought to determine whether living in a heavily industrial area would be a risk factor for respiratory symptoms [23]. They observed that people living closest to areas of high levels of SO_2_ had elevated incidences of sputum production and the presence of chronic cough. This study illustrates the relationship between sulfur dioxide levels and industrialization. Exposure to high concentrations of SO_2_ caused significant epithelial damage, and acute or chronic bronchitis with predominantly neutrophilic inflammation. Epidemiological studies have demonstrated the association between air pollution by SO_2_ and increased morbidity and exacerbations of asthma from aggravation of airway inflammation, induction of bronchospasm, and worsening of airway obstruction in asthma [24].

The daily number of hospitalizations for asthma was significantly positively associated with the NO_2_ concentration in our study, a finding consistent with previous reports. In the review by Zheng et al. [6] of 87 studies mentioned above, 66 studies showed a statistically significant correlation with NO_2_. Furthermore, in a recent review of 22 studies [19], NO_2_ was found to have a significant association with asthma exacerbation in children.

In our study, aged-stratified analysis showed that the association between air pollution and childhood asthma hospitalization differs with age. The daily numbers of hospitalizations due to asthma and the RRs for asthma hospitalization associated with air pollutants were higher among children aged 0–5 years than among children aged 6–18 years. In the largest Brazilian metropolis study, an increase of 1.4% in hospitalizations for total respiratory diseases was observed for each increase of 10 μg/m^3^ in the level of PM_10_, and in children younger than five, the effect was slightly higher, with a 1.9% increase in hospitalizations [25].

Air pollutants have many effects on the health of both adults and children, but the vulnerability of children is unique [26]. Children are more likely to be sensitive at a young age [27], a plausible interpretation being that children harbor immature lung growth [6], because only 80% of the alveoli in the lungs are formed after birth, and the lungs continue to change and develop through adolescence. The lungs of very young children are highly vulnerable to damage [28].

In the present study, we also found that the RRs for asthma hospitalization were slightly higher among males than females, a result consistent with previous studies performed in Athens, New York, Texas, Toyama (Japan) and the Basque region of Spain [12,28,29,30,31]. Samoli et al. [12] reported that adverse health effects of air pollution on childhood asthma were evident only in males. Epidemiologic studies of the effects of air pollution on respiratory health demonstrated significant differences by gender, and a review study of children suggested stronger effects among boys in early life and among girls in later childhood, which may vary by life stage, exposure, and hormonal status [32]. Males may also have more exposure to air pollution due to their activity patterns [12].

In light of the differing effects of air pollution in different regions, we also identified the greatest differences in RRs between different air pollution regions. The level of air pollution in the Kao-Ping region was higher than in other regions, the RR of asthma hospitalization was significantly higher than other regions. The level of air pollution in the rural region of Hua-Tung was lower, no association was observed in the rural region of Hua-Tung.

The current study had recognizable strengths and limitations. The major strengths of this study were that, to our knowledge, this was the first study to investigate childhood asthma hospitalization in different regions on a national scale, employing national-level hospitalization data over a duration of 12 years using big data analysis. Additionally, we stratified the results by air pollution region, gender, and age. However, there were some limitations of our study. There existed exposure measurement bias, as patients’ addresses were not available from the database, and we therefore assumed that a patient’s area of residence was close to the location of the hospital to which they were admitted. We employed the air pollutant concentrations measured at the monitoring station closest to the hospital to which a patient was admitted as a proxy of personal exposure, and thus these data did not represent the actual exposure of children with asthma. A series of studies suggested that risk estimates based on fixed-site ambient air pollution measurements are smaller than those estimated using personal measures. A study suggested that the actual exposure concentration be measured using personal devices [33]. Second, the presence of asthma was ascertained based on the diagnostic code obtained from the NHIRD, and hence there was the potential for differences in diagnostic measures; in addition, distinguishing asthma from other respiratory illnesses, such as wheezing or bronchitis, is particularly difficult in young children [34]. Another limitation was that we examined the associations between asthma hospitalization in children and air pollution levels in regions of differing air pollution, and it is necessary to identify other region-specific environmental factors and regional characteristics, such as topography and weather patterns, that could trigger asthma exacerbation in future studies.

## 5. Conclusions

Our findings showed significant short-term effects of ambient air pollutants on childhood asthma hospitalization, and SO_2_ was most strongly associated with daily asthma hospitalizations in children. The relative risks for asthma hospitalization associated with air pollutants were higher among children aged 0–5 than among children aged 6–18 and were higher among males than females. The effects of air pollution on childhood asthma were greater in the higher-level air pollution regions, while no association was observed in the lower-level air pollution rural regions.

The results of this study may prove important for healthcare providers and policymakers involved in developing region-specific approaches to the management of and medical resource allocation for asthma. Further studies can assess the threshold value for the air-pollutants at which they begin to affect health of children and health policy-making can benefit from more effective use of the research.

## Figures and Tables

**Figure 1 ijerph-16-00203-f001:**
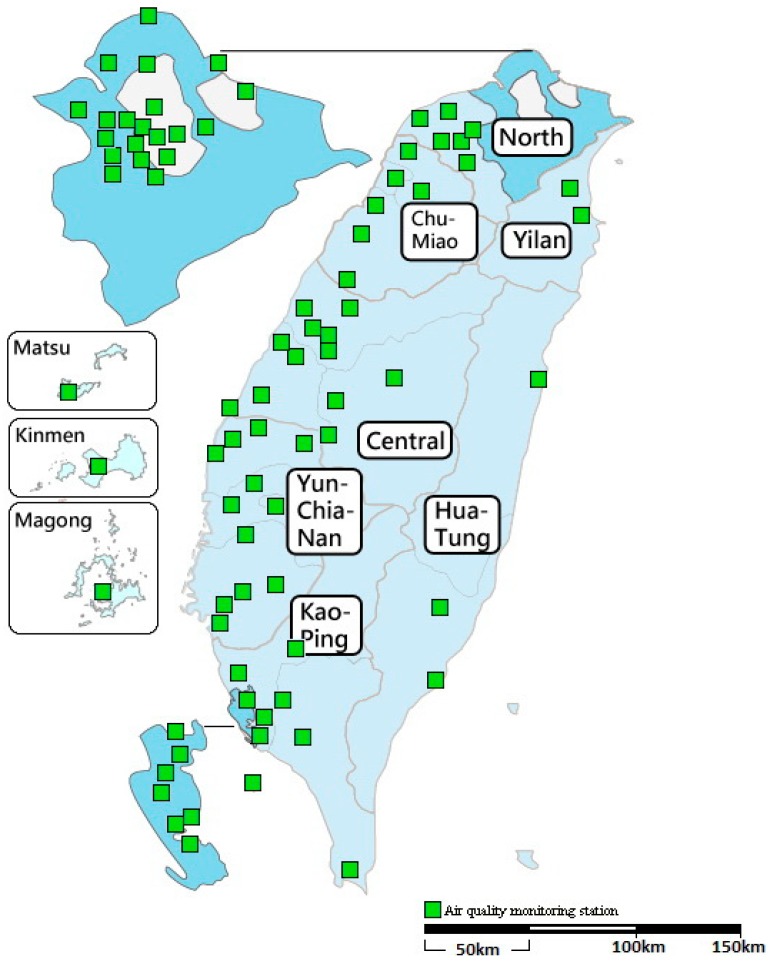
The seven air quality regions and an outlying islands region of Taiwan.

**Table 1 ijerph-16-00203-t001:** Characteristics of asthma hospitalization in different regions in Taiwan, 2001–2012.

Region	Northern		Chu-Miao		Central		Yun-Chia-Nan		Kao-Ping		Yilan		Hua-Tung		Outlying Islands		Total	
	*n*	%	*n*	%	*n*	%	*n*	%	*n*	%	*n*	%	*n*	%	*n*	%	*n*	%
Total subjects	22,482	100	4105	100	11,222	100	8608	100	8657	100	2387	100	1410	100	329	100	59,204	100
Age (years)																		
0–2	4177	18.58	819	19.95	1833	16.33	2309	26.82	1403	16.21	396	16.59	393	27.87	51	15.50	11,382	19.23
3–5	10,326	45.93	1656	40.34	4526	40.33	3497	40.63	3515	40.60	892	37.37	473	33.55	158	48.02	25,043	42.30
6–18	7979	35.49	1630	39.71	4863	43.33	2802	32.55	3739	43.19	1099	46.04	544	38.58	120	36.47	22,779	38.48
Gender																		
Male	14,588	64.89	2529	61.61	7275	64.83	5431	63.09	5398	62.35	1508	63.18	862	61.13	232	70.52	37,825	63.89
Female	7894	35.11	1576	38.39	3947	35.17	3177	36.91	3259	37.65	879	36.82	548	38.87	97	29.48	21,379	36.11
Season																		
Spring	5951	26.47	1108	26.99	3079	27.44	2383	27.68	2340	27.03	613	25.68	352	24.96	92	27.96	15,918	26.89
Summer	3986	17.73	780	19.00	2095	18.67	1656	19.24	1481	63.29	407	17.05	273	19.36	73	22.19	10,751	18.16
Autumn	4928	21.92	896	21.83	2522	22.47	1933	22.46	2069	23.90	534	22.37	333	23.62	83	25.23	13,298	22.46
Winter	7617	33.88	1321	32.18	3526	31.42	2636	30.62	2767	31.96	833	34.90	452	32.06	81	24.62	19,237	32.49
Mean length of stay (days)	3.54		4.25		3.68		3.55		4.23		3.79		4.51		3.89		3.93	

**Table 2 ijerph-16-00203-t002:** Summary of daily mean concentrations of air pollutants in Taiwan, 2001–2012.

				Percentiles				
Pollutants	Mean	SD	Minimum	25th	50th	75th	Maximum	*p*-Value *
**PM_2.5_ (μg/m^3^)**								
All regions	33.15	20.22	0.01	16.45	26.62	41.42	234.71	*p* < 0.001
Northern	27.21	15.53	0.59	14.96	23.13	33.46	206.21	
Chu-Miao	28.62	15.68	1.75	16.17	24.42	35.48	153.13	
Central	36.54	19.41	0.01	20.96	31.87	45.21	166.25	
Yun-Chia-Nan	38.33	20.24	1.25	21.46	33.54	48.04	234.71	
Kao-Ping	41.75	23.71	1	20.46	38.04	55.67	176.94	
Yilan	19.64	11.66	1.5	10.58	16.5	24.5	105.00	
Hua-Tung	16.43	9.55	1.67	8.96	13.62	19.88	90.50	
Outlying islands	32.08	21.15	1.33	15.47	26.25	41.08	174.21	
**PM_10_ (μg/m^3^)**								
All regions	57.82	33.51	1	30.96	47.33	71	772.71	*p* < 0.001
Northern	48.94	26.34	1	28.83	42.46	59.3	510.58	
Chu-Miao	48.41	25.61	1	29.04	42.33	59.05	486.75	
Central	61.61	31.83	1	36.5	54	76.58	512.00	
Yun-Chia-Nan	68.94	35.93	5.13	39.17	61.33	86.23	594.38	
Kao-Ping	70.33	38.94	1	35.88	62.16	94.08	615.96	
Yilan	39.46	20.57	4.17	24.83	34.33	47.36	538.46	
Hua-Tung	32.01	17.89	1	20.33	27.25	36.92	313.46	
Outlying islands	59.04	38.85	4.52	30.17	47.58	71.75	772.71	
**O_3_ (ppb)**								
All regions	28.83	12.47	0.1	18.5	26.1	34.7	107.78	*p* < 0.001
Northern	27.39	12.03	0.37	17.72	25.11	33.34	97.65	
Chu-Miao	28.89	10.98	0.5	20.01	27.04	34.84	84.18	
Central	27.49	11.09	0.13	18.45	25.52	33.32	83.12	
Yun-Chia-Nan	30.62	11.68	1.54	20.75	28.79	37.1	85.07	
Kao-Ping	30.02	13.78	0.13	17.9	27.69	38.47	93.83	
Yilan	25.99	10.54	0.62	17.66	24.86	31.98	78.52	
Hua-Tung	25.01	10.35	2.24	16.07	23.5	30.91	78.29	
Outlying islands	41.72	15.52	3.12	28.52	40.24	50.86	107.78	
**SO_2_ (ppb)**								
All regions	4.3	3.33	0.1	2.1	3.29	5	103.71	*p* < 0.001
Northern	4.23	2.85	0.1	2.12	3.47	5.18	69.16	
Chu-Miao	3.39	2.04	0.1	2	3.01	4.1	36.98	
Central	3.32	1.85	0.1	2.01	2.91	4.03	68.58	
Yun-Chia-Nan	3.66	1.94	0.1	2.32	3.27	4.37	43.23	
Kao-Ping	6.39	4.49	0.1	2.87	5.16	7.98	53.43	
Yilan	2.19	1.42	0.1	1.24	1.9	2.63	22.03	
Hua-Tung	1.67	1.13	0.1	0.88	1.61	2.12	41.48	
Outlying islands	4.18	3.53	0.1	1.67	2.73	5.2	34.13	
**NO_2_ (ppb)**								
All regions	18.13	10.43	0.01	9.54	15.53	22.71	116.2	*p* < 0.001
Northern	21.75	11.55	0.03	12.2	19.63	27.12	116.2	
Chu-Miao	15.44	6.29	0.28	10.62	13.98	18.5	51.4	
Central	18	8.54	0.01	10.9	16.03	22.21	76.03	
Yun-Chia-Nan	14.86	7.06	0.22	9.26	13.2	18.17	64.65	
Kao-Ping	19.75	12	0.02	10.05	17.54	25.75	103.77	
Yilan	11.22	3.96	0.84	8.01	10.42	13.15	27.76	
Hua- Tung	8.36	4.63	0.59	4.91	6.9	10	73.87	
Outlying islands	7.37	5.23	0.05	3.52	5.49	8.95	44.41	

* *p*-value for ANOVA comparing the mean concentrations of ambient air pollutants for each region. PM_2.5_: particulate matter ≤ 2.5 μm.PM_10_: particulate matter ≤ 10 μm. O_3_: ozone. SO_2_: sulfur dioxide. NO_2_: nitrogen dioxide.

**Table 3 ijerph-16-00203-t003:** Relative risks of asthma hospitalization associated with different air pollutants, single pollutant model vs. multiple pollutants model. CI: confidence interval; RR: relative risk.

	0–18 Years Old					
	Single Pollutant Model			Multiple Pollutants Model		
Pollutants	RR	95% CI	*p*-Value	RR	95% CI	*p*-Value
PM_2.5_	1.156 ***	1.142–1.170	0.000	0.968 **	0.948–0.988	0.002
PM_10_	1.120 ***	1.107–1.134	0.000	1.028 *	1.005–1.050	0.015
O_3_	0.969 ***	0.957–0.981	0.000	0.950 ***	0.937–0.964	0.000
SO_2_	1.367 ***	1.349–1.385	0.000	1.537 ***	1.507–1.566	0.000
NO_2_	1.065 ***	1.053–1.076	0.000	0.862 ***	0.849–0.874	0.000

* *p*-value < 0.05, ** *p*-value < 0.01, *** *p*-value < 0.001.

**Table 4 ijerph-16-00203-t004:** Relative risk of asthma hospitalization based on air pollutants, stratified by age and gender. Single pollutant model vs. multiple pollutants model.

	**Single Pollutant Model**					
	**0–5 Years Old**			**6–18 Years Old**		
**Pollutants**	**RR**	**95% CI**	***p*-Value**	**RR**	**95% CI**	***p*-Value**
PM_2.5_	1.190 ***	1.172–1.208	0.000	1.088 ***	1.067–1.109	0.000
PM_10_	1.123 ***	1.106–1.140	0.000	1.100 ***	1.078–1.121	0.000
O_3_	0.962 ***	0.947–0.977	0.000	0.977 *	0.958–0.997	0.025
SO_2_	1.428 ***	1.405–1.452	0.000	1.245 ***	1.219–1.271	0.000
NO_2_	1.060 ***	1.045–1.075	0.000	1.064 ***	1.045–1.082	0.000
	**Multiple Pollutants Model**					
	**0–5 Years Old**			**6–18 Years Old**		
**Pollutants**	**RR**	**95% CI**	***p*-Value**	**RR**	**95% CI**	***p*-Value**
PM_2.5_	1.008	0.982–1.035	0.526	0.909 ***	0.879–0.939	0.000
PM_10_	0.987	0.958–1.016	0.369	1.083 ***	1.047–1.119	0.000
O_3_	0.940 ***	0.924–0.957	0.000	0.967 **	0.945–0.989	0.003
SO_2_	1.647 ***	1.607–1.689	0.000	1.346 ***	1.306–1.387	0.000
NO_2_	0.827 ***	0.811–0.843	0.000	0.924 ***	0.902–0.946	0.000
	**Single Pollutant Model**					
	**Male**			**Female**		
**Pollutants**	**RR**	**95% CI**	***p*-Value**	**RR**	**95% CI**	***p*-Value**
PM_2.5_	1.151 ***	1.133–1.168	0.000	1.143 ***	1.121–1.166	0.000
PM_10_	1.115 ***	1.098–1.131	0.000	1.113 ***	1.091–1.136	0.000
O_3_	0.965 ***	0.950–0.980	0.000	0.968 **	0.968–0.988	0.002
SO_2_	1.359 ***	1.337–1.381	0.000	1.332 ***	1.304–1.361	0.000
NO_2_	1.063 ***	1.049–1.078	0.000	1.051 ***	1.032–1.070	0.000
	**Multiple Pollutants Model**					
	**Male**			**Female**		
**Pollutants**	**RR**	**95% CI**	***p*-Value**	**RR**	**95% CI**	***p*-Value**
PM_2.5_	0.970 *	0.945–0.995	0.020	0.969	0.936–1.002	0.067
PM_10_	1.023	0.995–1.052	0.099	1.037 *	1.000–1.075	0.048
O_3_	0.947 ***	0.931–0.964	0.000	0.947 ***	0.925–0.970	0.000
SO_2_	1.525 ***	1.489–1.562	0.000	1.495 ***	1.448–1.543	0.000
NO_2_	0.865 ***	0.849–0.881	0.000	0.855 ***	0.834–0.876	0.000

* *p*-value <0.05, ** *p*-value < 0.01, *** *p*-value < 0.001.

**Table 5 ijerph-16-00203-t005:** Relative risks of asthma hospitalization associated with air pollutants in different regions—a single pollutant model.

	**Single Pollutant Model**					
	**Northern**			**Chu-Miao**		
**Pollutants**	**RR**	**95%CI**	***p*-Value**	**RR**	**95%CI**	***p*-Value**
PM_2.5_	1.075 ***	1.057–1.093	0.000	1.022	0.954–1.093	0.539
PM_10_	1.038 ***	1.021–1.054	0.000	1.020	0.953–1.090	0.560
O_3_	0.989	0.970–1.008	0.237	0.981	0.911–1.055	0.598
SO_2_	1.109 ***	1.093–1.126	0.000	1.049	0.979–1.124	0.175
NO_2_	0.995	0.977–1.014	0.610	1.022	0.964–1.082	0.459
	**Central Taiwan**			**Yun-Chia-Nan**		
**Pollutants**	**RR**	**95%CI**	***p*-Value**	**RR**	**95%CI**	***p*-Value**
PM_2.5_	1.026 *	1.001–1.052	0.043	1.049 *	1.005–1.094	0.027
PM_10_	1.085 ***	1.058–1.113	0.000	1.054 *	1.008–1.099	0.018
O_3_	0.952 ***	0.926–0.979	0.000	0.992	0.951–1.0034	0.689
SO_2_	1.216 ***	1.182–1.251	0.000	1.152 ***	1.094–1.213	0.000
NO_2_	1.011	0.985–1.037	0.424	1.069 ***	1.030–1.110	0.000
	**Kao-Ping**			**Yilan**		
**Pollutants**	**RR**	**95%CI**	***p*-Value**	**RR**	**95%CI**	***p*-Value**
PM_2.5_	1.083 ***	1.045–1.123	0.000	0.995	0.924–1.070	0.899
PM_10_	1.096 ***	1.053–1.141	0.000	0.980	0.918–1.039	0.529
O_3_	1.033	0.995–1.073	0.091	0.956	0.884–1.033	0.255
SO_2_	1.185 ***	1.129–1.244	0.000	1.070	0.987–1.160	0.101
NO_2_	1.079 ***	1.044–1115	0.000	1.025	0.964–1.088	0.416
	**Hua-Tung**			**Outlying Islands**		
**Pollutants**	**RR**	**95%CI**	***p*-Value**	**RR**	**95%CI**	***p*-Value**
PM_2.5_	0.986	0.901–1.077	0.764	1.026	0.849–1.23	0.784
PM_10_	1.023	0.936–1.115	0.603	1.030	0.834–1.257	0.780
O_3_	1.019	0.928–1.118	0.686	1.079	0.868–1.340	0.492
SO_2_	0.977	0.882–1.080	0.652	1.023	0.857–1.210	0.794
NO_2_	1.005	0.931–1.082	0.900	1.049	0.825–1.320	0.690

* *p*-value < 0.05, ** *p*-value < 0.01, *** *p*-value < 0.001.

**Table 6 ijerph-16-00203-t006:** Relative risks of asthma hospitalization associated with air pollutants in different regions—a multiple pollutants model.

	**Multiple Pollutants Model**					
	**Northern**			**Chu-Miao**		
**Pollutants**	**RR**	**95% CI**	***p*-Value**	**RR**	**95% CI**	***p*-Value**
PM_2.5_	1.076 ***	1.046–1.107	0.000	0.981	0.787–1.227	0.867
PM_10_	0.977	0.949–1.006	0.126	1.003	0.812–1.228	0.980
O_3_	0.965 ***	0.946–0.986	0.000	0.993	0.914–1.077	0.859
SO_2_	1.193 ***	1.167–1.219	0.000	1.054	0.956–1.161	0.289
NO_2_	0.833 ***	0.809–0.856	0.000	1.009	0.935–1.088	0.807
	**Central Taiwan**			**Yun-Chia-Nan**		
**Pollutants**	**RR**	**95% CI**	***p*-Value**	**RR**	**95% CI**	***p*-Value**
PM_2.5_	0.860 ***	0.822–0.901	0.054	0.917	0.839–1.003	0.059
PM_10_	1.121 ***	1.068–1.174	0.189	1.051	0.971–1.131	0.201
O_3_	0.989	0.959–1.019	0.919	0.988	0.941–1.037	0.621
SO_2_	1.347 ***	1.292–1.404	0.000	1.178 ***	1.092–1.271	0.000
NO_2_	0.888 ***	0.857–0.920	0.143	1.023	0.973–1.075	0.377
		**Kao-Ping**			**Yilan**	
**Pollutants**	**RR**	**95% CI**	***p*-Value**	**RR**	**95% CI**	***p*-Value**
PM_2.5_	0.992	0.933–1.055	0.794	1.023	0.881–1.192	0.767
PM_10_	1.004	0.933–1.078	0.914	0.944	0.826–1.055	0.355
O_3_	1.015	0.972–1.060	0.490	0.978	0.894–1.071	0.637
SO_2_	1.172 ***	1.086–1.265	0.000	1.073	0.973–1.184	0.160
NO_2_	1.014	0.972–1.058	0.512	1.015	0.947–1.086	0.671
	**Hua-Tung**			**Outlying islands**		
**Pollutants**	**RR**	**95% CI**	***p*-Value**	**RR**	**95% CI**	***p*-Value**
PM_2.5_	0.914	0.791–1.059	0.231	0.957	0.605–1.51	0.850
PM_10_	1.082	0.945–1.226	0.236	0.992	0.609–1.533	0.973
O_3_	1.032	0.931-1.143	0.550	1.098	0.844-1.423	0.482
SO_2_	0.972	0.873–1.079	0.598	1.021	0.805–1.279	0.857
NO_2_	1.019	0.935–1.109	0.662	1.051	0.757–1.424	0.760

*** *p*-value < 0.001.

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
