# Peer review of "The Short-Term Effects of Ambient Air Pollutants on Childhood Asthma Hospitalization in Taiwan: A National Study"

_ijerph, 2019, doi:10.3390/ijerph16020203_

Reviewer 1 Report

Dear authors,

I found your article very interesting, however I have some comments on it.

Author Response

December 30, 2018

Dear Reviewer,

Attached please find the revised manuscript ijerph-401877 entitled “The short-term effects of ambient air pollutants on childhood asthma hospitalization in regions with differing air pollution in Taiwan: A national study” for your consideration to be published in IJERPH. The title has been recommended to be shortened by the first reviewer as “The short-term effects of ambient air pollutants on childhood asthma hospitalization in Taiwan: A national study”. This manuscript was revised point by point carefully according to all valuable suggestions from reviewers. We believe that the findings of this study will appeal to readers interested in public concern. We’d like to thank the editorial team in advance for taking the time to review our manuscript to improve the quality of the paper. Our responses to you are as the following.

Yours sincerely,

Chien-Lung Chan

E-mail: clchan@saturn.yzu.edu.tw

Author's Reply to the Review Report (Reviewer 1)

1.     Title – can be shortened – it’s clear that authors describe influence of the air pollutants on childhood asthma – then different regions will be analyzed.

Ans. Many thanks for the valuable suggestion

We revised the title as “The short-term effects of ambient air pollutants on childhood asthma hospitalization in Taiwan: A national study”.

2.     Line 41-43 – all phrase should be divided to two - after …causation…. I suggest inserting a dot

Ans.

We inserted a dot and revised the part.

3.     Line 62 – abbreviation DD?

Ans.

The NHIRD includes various data subsets, such as inpatient expenditure by admission (DD), details of inpatient orders (DO), ambulatory care expenditure by visit (CD), and details of ambulatory care orders (OO). In this study, we used the inpatient expenditure by admission DD data subset. We added more description in  section “2.1. Asthma hospitalization data”

.

4. Line 72 – 79 – detailed describing of the regions is not necessary for readers – all analysis showed only regions not cities, beside of that there is some differences – line 78 and data on the map presented in Fig. 1. Lianjiang County =LienChiang County? Jinmene Count=KinMen County?

Also, important that presented data should be describe as ….was …. Stations. Actual data are not the same e.g. Jiu-Miao region – 6 stations not 7 as authors describe – data from web site taqm.epa.gov.tw

Fig. 1 – Hsinchu City is invisible, for the clarification - 8 should be also putted to the LienChiang and KinMen

Ans. We would like to thank the reviewer for the valuable comment.

We deleted the detail descriptions of the regions and revised Figure 1 according to the EPA. Jiu-Miao region has 6 stations and we revised.

5.Line 123-131 – all data are presented in the table 2 – it’s not necessary to duplicate them

Ans. We would like to thank the reviewer for the valuable comment.

 We deleted them.

6.Table 2 – p-value refer to ???? e.e PM2.5 – Kao-Ping vs. all regions? Etc.

Ans.

The p-value showed the significance of difference mean concentrations of ambient air pollutants for each region by ANOVA test. We added a supplementary note for the p-value on Table 2.

7.Table 3 – legend is necessary – explain all stars used in the text

For the clarification table 3 should be divided into two – one from the whole group of children, second – with single and multiple pollutant model in coincidence to the age

Also, the same objection – authors should choose how to present data – in the text or in the table – table 3 and table 4. For the more visibility, I strongly recommend removing from the text specific statistical data – they only obscure the description of the differences.

Ans.

We have added a supplementary note for the tables 3, 4, 5, 6 with *p-value < 0.05, ** p-value < 0.01, *** p-value < 0.001

We divided the Table 3 into 2 tables (Table 3 and Table 4) and removed the statistical data from the texts.

8.Authors should describe what is sulfur dioxide as they describe (line 208-213) other particles – why they are so important for our quality of life, especially in the context of asthma and asthma exacerbation

Line 226 – authors found that SO2 was the most important pollutant which was strongly associated with daily asthma hospitalization – for readers please explain why, describe mechanism of action of SO2.

I found study by Hwang and col. Traffic related air pollution as a determinant of asthma among Taiwanese school children – this data for me should be presented by authors in the discussion.

Ans. Many thanks for the valuable suggestion.

We added more description the mechanism of SO2 in asthma exacerbation as follows:

“The main sources of SO2 in the developed world are primary emissions during energy production or industrial processes [19]. SO2 is a recognized environmental toxicant that can act to promote airway responses in a concentration-dependent manner, possibly through its ability to induce local oxidative stress[21]. A high probability of SO2 exposure may be confined to the factory area itself and within the vicinity of several square miles or the original site of its generation [22]. A study conducted in Russia by Nieminen et al. (2013) sought to determine whether living in a heavily industrial area would be a risk factor for respiratory symptoms [23]. They observed that people living closest to areas of high levels of SO2 had elevated incidences of sputum production and the presence of chronic cough. This study illustrates the relationship between sulfur dioxide levels and industrialization. Exposure to high concentration of SO2 caused significant epithelial damage, acute or chronic bronchitis with a predominantly neutrophilic inflammation. Epidemiological studies have demonstrated the association between air pollution by SO2 and increased morbidity and exacerbations of asthma from aggravation of airway inflammation, induction of bronchospasm and worsening of airway obstruction in asthma[20]”.

We cited Hwang and colleagues’ study ”Traffic related air pollution as a determinant of asthma among Taiwanese school children” as the reviewer’s suggestion and added more description at “discussion” about the effect of O3 and SO2 on childhood asthma as follows:.

“In nationwide cross sectional study about the effect of air pollutants on the risk of asthma among school children in 2001 in Taiwan[17], Hwang et al. found the risk of childhood asthma was positively associated with O3 (adjusted OR 1.138, 95% confidence interval 1.001 to 1.293).”

“However, the risk of childhood asthma was not related to SO2 (adjusted OR 0.874, 95% CI 0.729 to 1.054) in 2001 in Taiwan by Hwang et al.[17]. Because urban air pollution constitutes a complex mixture of several compounds, SO2 and PM10 concentrations were also correlated (Hwang et al.), assessment of the independent effects of different pollutants is difficult [17]”.

9.Conclusions,  for me, should be a little be changed:

“Reductions in the concentrations of pollutants could result in decreased numbers of hospitalizations” – actual data shown that quality of air in Taiwan is quite good (only some regions especially in the south part have “unhealthy air”)

“The results of this study may prove important for healthcare providers and policymakers involved in developing region-specific approaches to the management of and medical source allocation for asthma and provide support for the government towards implementing policies to reduce the levels of air pollution, especially SO2 “– authors should describe actual data from the government. Between 2012 – 2018 as I hope, especially in this region, government implement  policies to reduce the levels of the air pollution.

Ans. We would like to thank the reviewer for the valuable comment.

We deleted the parts and revised the conclusion as “The results of this study may prove important for healthcare providers and policymakers involved in developing region-specific approaches to the management of and medical resource allocation for asthma. Further studies can assess the threshold values for the air-pollutants at which they begin to affect health of children and health policy-making can benefit from more effective use of the research.”

Reviewer 2 Report

This is a significant study on effects of air pollutants on public health in Taiwan. Similar studies have been done previously for different cities and countries. For the most part, conclusions of this study are consistent with previously published studies. However, the negative association between O3 and childhood asthma hospitalization is surprising.

Specific Comments:

1.      There is enough introduction to Asthma but not enough introduction to air-pollutants discussed in the study. Adding this information will be useful for readers.

2.      Is it possible to add the air quality monitoring stations (page 2) to the map in Figure 1? This information will be easy to visualize on the map and get a better understanding of data collection for each region.

3.      Table 1 is confusing and needs legends for what the %values mean. For instance, the values in first row (0-18 total, (n)) is % values of total hospitalizations whereas values from 2nd row onwards lists %values of hospitalizations within each region.

4.      % hospitalization for each region can be misleading without knowing the population of children from that region.

5.      Authors need to state a reason why humidity and temperature were removed from analysis. Both humidity and temperature can affect air-pollutant triggered asthma symptoms. What would the data (Table 3) look like if these factors were counted in?

6.      Tables with stars (*) need legends to explain them. P-values are more informative in legends than the main text.

7.      Line 195: “import” to “important”

8.      Considering the wealth of this data, is it possible to determine a threshold value for the air-pollutants at which they begin to affect health of children. Policymakers may benefit from this information.

Author Response

December 30, 2018

Dear Reviewer,

Attached please find the revised manuscript ijerph-401877 entitled “The short-term effects of ambient air pollutants on childhood asthma hospitalization in regions with differing air pollution in Taiwan: A national study” for your consideration to be published in IJERPH. This manuscript was revised point by point carefully according to all valuable suggestions from reviewers. We believe that the findings of this study will appeal to readers interested in public concern. We’d like to thank the editorial team in advance for taking the time to review our manuscript to improve the quality of the paper. Our responses to you are as the following.

Yours sincerely,

Chien-Lung Chan

E-mail: clchan@saturn.yzu.edu.tw

Author's Reply to the Review Report (Reviewer 2)

1. There is enough introduction to Asthma but not enough introduction to air-pollutants discussed in the study. Adding this information will be useful for readers.

 Ans. Many thanks for the valuable suggestion

We added more description at “introduction” about air pollution contributes to adverse health effects as the following.

“Air pollution and its related health impact have become a major concern over the past few years. Ambient air pollution accounts for an estimated 4.2 million deaths per year due to stroke, heart disease, lung cancer and chronic respiratory diseases. 91% of the world’s population lives in places where air quality exceeds WHO guideline limits. The pollutants with the strongest evidence of health effects are particulate matter (PM), O3, NO2 and SO2 [4]. Ambient air pollution levels have been found to be associated with hospitalization due to asthma [5]. A systematic review and meta-analysis of the association between air pollutants and asthma  hospital admissions showed that air pollutants were associated with a significant increased risk of asthma hospitalizations [6].”

2. Is it possible to add the air quality monitoring stations (page 2) to the map in Figure 1? This information will be easy to visualize on the map and get a better understanding of data collection for each region.

 Ans. Many thanks for the valuable suggestion

We added the air quality monitoring stations to the map and revised Figure 1 according to the EPA

3.  Table 1 is confusing and needs legends for what the % values mean. For instance, the values in first row (0-18 total, (n)) is % values of total hospitalizations whereas values from 2nd row onwards lists %values of hospitalizations within each region.

 Ans. We would like to thank the reviewer for the valuable comment. We revised table 1

Table 1. Characteristics of asthma hospitalization in different regions in Taiwan, 2001–2012

Region

Northern

Jhu-Miao

Central

Yun-Chia-Nan

Kao-Ping

Yilan

Hua-Dong

Outlying islands

Total

N

%

N

%

N

%

N

%

N

%

N

%

N

%

N

%

N

%

Total subjects

22482

100

4105

100

11222

100

8608

100

8657

100

2387

100

1410

100

329

100

59204

100

        Age   (year)

0–2

4177

18.58

819

19.95

1833

16.33

2309

26.82

1403

16.21

396

16.59

393

27.87

51

15.50

11382

19.23

3–5

10326

45.93

1656

40.34

4526

40.33

3497

40.63

3515

40.60

892

37.37

473

33.55

158

48.02

25043

42.30

6–18

7979

35.49

1630

39.71

4863

43.33

2802

32.55

3739

43.19

1099

46.04

544

38.58

120

36.47

22779

38.48

        Gender

Male

14588

64.89

2529

61.61

7275

64.83

5431

63.09

5398

62.35

1508

63.18

862

61.13

232

70.52

37825

63.89

Female

7894

35.11

1576

38.39

3947

35.17

3177

36.91

3259

37.65

879

36.82

548

38.87

97

29.48

21379

36.11

        Season

Spring

5951

26.47

1108

26.99

3079

27.44

2383

27.68

2340

27.03

613

25.68

352

24.96

92

27.96

15918

26.89

Summer

3986

17.73

780

19.00

2095

18.67

1656

19.24

1481

63.29

407

17.05

273

19.36

73

22.19

10751

18.16

Autumn

4928

21.92

896

21.83

2522

22.47

1933

22.46

2069

23.90

534

22.37

333

23.62

83

25.23

13298

22.46

winter

7617

33.88

1321

32.18

3526

31.42

2636

30.62

2767

31.96

833

34.90

452

32.06

81

24.62

19237

32.49

Mean length of stay (day)

3.54

4.25

3.68

3.55

4.23

3.79

4.51

3.89

3.93

4. % hospitalization for each region can be misleading without knowing the population of children from that region.

 Ans. We would like to thank the reviewer for the valuable comment. We revised table 1

5. Authors need to state a reason why humidity and temperature were removed from analysis. Both humidity and temperature can affect air-pollutant triggered asthma symptoms. What would the data (Table 3) look like if these factors were counted in?

Ans. We would like to thank the reviewer for the valuable comment.

The specific aim for this study was to assess the effects of environmental air pollution on childhood asthma hospitalization. It is well documented that temperature and humidity can be potential confounders by other studies, so the ambient daily temperature and relative humidity were used to control for meteorological conditions in our study. In the result, Table 3 had summarized the relative risk of hospitalization (RR) due to asthma and air pollution with a change in the interquartile range (IQR) of the level of the pollutant exposed to after adjusting for the meteorological factors of temperature and humidity. We described more description in section 2.2 Air pollution and meteorological data, “It is well documented that temperature and humidity can be potential confounders by other studies, so the ambient daily temperature and relative humidity were used to control for meteorological conditions. Meteorological data including the ambient daily average temperature and relative humidity were obtained from the Central Weather Bureau”.

6. Tables with stars (*) need legends to explain them. P-values are more informative in legends than the main text.

Ans. Many thanks for the valuable suggestion.

We have added a supplementary note for the tables 3, 4, 5, 6 with *p-value < 0.05, ** p-value < 0.01, *** p-value < 0.001.

7. Line 195: “import” to “important”

Ans. Many thanks for the valuable suggestion.

We revised it.

8. Considering the wealth of this data, is it possible to determine a threshold value for the air-pollutants at which they begin to affect health of children. Policymakers may benefit from this information.

Ans. Many thanks for the valuable suggestion.

We added your suggestion in our conclusions and revised our conclusion. We will analysis the threshold value for the air-pollutants at which they begin to affect health of children in the future study. We revised the conclusion as “The results of this study may prove important for healthcare providers and policymakers involved in developing region-specific approaches to the management of and medical resource allocation for asthma. Further studies can assess the threshold value for the air-pollutants at which they begin to affect health of children and health policy-making can benefit from more effective use of the research.”